# Physicochemical Properties of Black Korean Goat Meat with Various Slaughter Ages

**DOI:** 10.3390/ani13040692

**Published:** 2023-02-16

**Authors:** Da-Mi Choi, Kyu-Min Kang, Sun-Moon Kang, Hack-Youn Kim

**Affiliations:** 1Department of Animal Resources Science, Kongju National University, Yesan 32439, Republic of Korea; 2Animal Products Utilization Division, National Institute of Animal Science, Rural Development Administration, Wanju-gun 55365, Republic of Korea

**Keywords:** black goat, slaughter age, physicochemical properties

## Abstract

**Simple Summary:**

As the aging population is rapidly increasing worldwide, consumer interest in age-friendly foods and healthy foods is increasing. Recently, as the nutritional value of black goat meat has been proven, the consumption of black goats has increased. However, the best age of slaughter has not been established for black goats, and studies on meat characteristics according to the age of slaughter are also insufficient. Therefore, in this study, the physicochemical properties of black goat meat according to the slaughter age (3, 6, 9, 12, 24, and 36 months of age) were investigated. As the age of the black goats increased, the shear force, redness, and yellowness increased, and the brightness decreased. In addition, the results for water holding capacity, cooking yield, mineral content, and fatty acid composition, showed that it is appropriate to slaughter black goats after 12 months of age.

**Abstract:**

This study was conducted to analyze the physicochemical properties of black goat meat according to the slaughter age (3, 6, 9, 12, 24, 36 months). The moisture content tended to decrease, whereas the fat content, pH, and free amino acid composition tended to increase with increasing slaughter age. The collagen content increased significantly with the increasing slaughter age (*p* < 0.05). The cooking yield showed a tendency to increase up to 12 months of age, and there was no significant difference after 12 months of age. In all mineral contents, the sample for 12 months of age showed higher values than the others. Considering fatty acid composition, the saturated fatty acid content of the 12-month sample had a lower value than the other samples. However, the unsaturated fatty acid of the 12-month sample had higher values than the other samples. Therefore, the best slaughter age for black goats occurs at 12 months of age when nutrition is excellent.

## 1. Introduction

The black goat (*Capra hircus*) belongs to the Bovidae, is a domesticated wild goat, and is characterized by its small size and black fur [1]. Because the black goat has a unique odor, it has been mainly consumed for medicinal purposes in the form of extracts rather than meat in South Korea [2]. Extracts are made by boiling a black goat with fat and connective tissue removed in water with added medicinal materials. In Western countries such as the United States and Europe, milk produced by black goats is used as processed foods such as black goat milk and cheese [3]. However, as its value as a low-fat, high-protein meat source is becoming noticed, the consumption patterns for black goat is also changing towards meat [4,5]. Black goat meat has high calcium, magnesium, and iron contents, and its essential amino acids and fatty acids are richer than pork and beef [6,7]. As a result, consumption of black goats has steadily increased, and in 2019, 542,744 black goat hybrids were being raised on 14,664 farms in Korea [8].

As the consumption of black goat meat increases, various studies on goats are being conducted to increase its utility as edible meat. Studies have been conducted on goat meat quality characteristics according to the live weight of goats during slaughter [9], the meat quality characteristics according to the feed supplied to the goats [10,11], and physicochemical properties according to the parts [12]. In addition, a study was conducted on the meat quality characteristics of Korean native black goats according to muscle type and age [13].

However, Korean domestic studies on the establishment of optimal slaughter age for black goats and studies on edible meat characteristics according to slaughter age are insufficient. Therefore, this study analyzed the physicochemical quality characteristics of black goat meat according to the slaughter age (3, 6, 9, 12, 24, 36 months) to identify the appropriate slaughter age and to utilize it as basic data for research on black goat meat products.

## 2. Materials and Methods

### 2.1. Sample Preparation

A total of 30 female black goats (Boer × black Korean goats; 3, 6, 9, 12, 24, and 36 months of age, five each) slaughtered at the Gaon (Gang-jin, Republic of Korea) were collected. The black goat was bred in a barn by feeding mixed pasture hay free-range diet and formulated feed (crude protein 17%, crude fat 3%, crude fiber 15%, crude ash 10%, ME (Metabolizable energy) 3.0 MCal/kg) at around 1% of body weight, and the black goat kids were weaned at around 3 months of age. The body weight and carcass yield by age of the black goats used in the experiment were 3 months age (10–15 kg; 7–8 kg), 6 months age (20–25 kg; 10–13 kg), 9 months age (30–35 kg; 15 kg–18 kg), 12 months age (45–50 kg; 21–26 kg), 24 months age (50–55 kg; 25–28 kg), and 36 months age (55–65 kg; 26–31 kg). The slaughtered black goat was cooled at 4 °C for 24 h and then transported to the laboratory while maintaining the condition at 0~5 °C. After that, the *longissimus lumborum* muscle was collected, excess fat and connective tissue were removed, and samples were stored at −24 °C for use.

### 2.2. Proximate Compositions

The moisture content, protein content, fat content, and collagen content of the samples were measured using Foodscanner (DA 6200, PerkinElmer, Waltham, MA, USA). Raw meat was homogenized prior to measurements, and samples were analyzed on multiple DA 6200 instruments using magnetic coupled plastic sample cups, 14 mm depth. This analysis method is based on the AOAC method and is based on transmission Diode Array Near-Infrared technology with the measurement light transmitted through the sample.

### 2.3. Shear Force

Shear force measurement samples (2.0 × 1.0 × 1.0 cm; length × width × height) were prepared by cutting the core of a black goat *longissimus lumborum*, that had been cooled after heating, in parallel to the muscle fiber direction. Then, using a texture analyzer (TA1, Lloyd, Largo, FL, USA) equipped with a V-blade, it was cut perpendicular to the muscle fiber direction. At this time, the analysis conditions were as follows: a test speed 2.0 mm/s, distance 22.0 mm, trigger force 5.0 g, and the measured values were expressed in Newtons (N).

### 2.4. Color

The color was measured using a colorimeter (CR-10, Minolta, Tokyo, Japan) equipped with a pulsed xenon lamp, 2° standard observer, aperture of 8 mm, and illuminant D65. Before measuring, the device was calibrated using a white standard plate (CLE L*: +97.83, CIE a*: −0.43, CIE b*: −1.98), and the lightness (CIE L*), redness (CIE a*), and yellowness (CIE b*) were recorded. The samples were bloomed at 25 °C for 30 min and then the color was measured.

### 2.5. pH at 24 h Post-Mortem (pH_24_)

The pH of the sample was measured by contacting the surface of the meat with a Flat Surface pH meter (8135BN, Thermo Fisher Scientific, Waltham, MA, USA) 24 h post-mortem. The pH meter was calibrated with pH 4.01, pH 7.00, and pH 10.00 buffer solution (Suntex Instruments Co, Ltd., Taipei, Taiwan).

### 2.6. Water Holding Capacity (WHC)

The WHC of the sample was measured by the Filter paper press method [14]. An amount of 0.3 g of the sample inner part was placed at the center of the filter paper (Whatman No. 1, GE Healthcare, Chicago, IL, USA) and compressed for 3 min under constant pressure using a plexiglass plate device. After that, the area of the sample and the area of the leaked water were measured using a digitizing area-lines sensor (MT-10S, MT Precision, Tokyo, Japan). The WHC was calculated as a percentage by substituting the following formula.
(1)$$\mathrm{WHC}\;(\%)=\frac{{\mathrm{Meat}\;\mathrm{area}\;(\mathrm{mm}}^{2})}{{\mathrm{Exudation}\;\mathrm{area}\;(\mathrm{mm}}^{2})}$$

### 2.7. Cooking Yield

The black goat *longissimus lumborum* was cut into steaks (6.0 × 6.0 × 2.5 cm; length × width × height). After that, the sample was cooked at 80 °C for 30 min using a chamber (10.10ESI/sk, Alto Shaam, Menomonee Falls, WI, USA), cooled at 25 °C for 1 h, and the cooking yield was determined by measuring the weight before cooking and after cooking of the sample. The measured value was calculated as a percentage by substituting the following formula.
(2)$$\mathrm{Cooking}\;\mathrm{yield}\;\left(\%\right)=\frac{\;\mathrm{Sample}\;\mathrm{weight}\;\mathrm{after}\;\mathrm{cooking}\;(g)}{\mathrm{Sample}\;\mathrm{weight}\;\mathrm{before}\;\mathrm{cooking}\;(g)}\times 100$$

### 2.8. Mineral

The mineral content was analyzed using the method of Kang et al. [15]. Take 500 mg of sample in the vessel, mix 7 mL of HNO_3_ and 2 mL of H_2_O_2_, pre-treat using microwave (Titan MPS Microwave, PerkinElmer, Waltham, MA, USA) for 10 min, and then dilute to 3.4% concentration with tertiary distilled water. For the sample, the concentration of minerals was calculated by measuring the intensity of the ion beam using Inductively Coupled Plasma Optical Emission Spectroscopy (ICP-OES, OPTIMA200 DV, PerkinElmer, Waltham, MA, USA). The analysis conditions are as follows: step 1 (temperature: 145 °C, pressure max: 70 min, ramp time: 5 min, hold time: 5 min, power: 60%), step 2 (temperature: 200 °C, pressure max: 80 min, ramp time: 5 min, hold time: 20 min, power: 90%), step 3 (temperature: 50 °C, pressure max: 80 min, ramp time: 1 min, hold time: 10 min, power: 0%). The standard products for quantification were multi-element calibration standards (ICP multi-element standard solution IV, Sigma–Aldrich Co., St. Louis, MO, USA), and the phosphorus, potassium, magnesium, calcium, and sodium were measured.

### 2.9. Free Amino Acid

The free amino acid content was measured using the method of Kang et al. [15]. Twenty grams of samples were mixed with 10 mL of 5% trichloroacetic acid and centrifuged at 4 °C for 15 min. Then, it was reacted in a 4 °C refrigerator for 1 h and impurities removed by filtering through a filter (PTFE 13 mm 0.2 μm, Advantec, Irvine, CA, USA). The filtered sample was analyzed and quantified using a liquid chromatography system (Ultimate™ WPS-3000RS, Thermo Fisher Scientific, Waltham, MA, USA) equipped with a reversed-phase column (ACCQ-TAG ULTRA C18 1.7 μm, Waters, Milford, MA, USA). The 10% Waters AccQ-tag Eluent A concentrate was used for mobile phase A, and th 100%Waters AccQ-tag Eluent B concentrate for B. The standard products for quantification were amino acid standard, L-glutamine, L-asparagine, L-tryptophan, γ-aminobutyric acid (Sigma–Aldrich Co., St. Louis, MO, USA), and the total amino acid content was quantified by measuring absorbance at 260 nm.

### 2.10. Fatty Acid Composition

For fatty acid composition, lipids were extracted using the method of Folch et al. [16]. The sample and chloroform-methanol (2:1) were added and homogenized for 1 min at 1296× *g* with a homogenizer (T25 Digital Ultra-Turrax, Ika Werke GmbH & Co., Staufen, Baden–Württemberg, Germany). After that, 0.88% KCl was added and centrifuged for 10 min using a centrifuge (Avanti J-E, Beckman Coulter, Fullerton, CA, USA) at 2 °C and 3000× *g* conditions. The supernatant was removed, and the lower layer was filtered with filter paper (Whatman International Ltd., Maidstone, Kent, England). Then, it was concentrated using an N_2_ gas blow concentrator (MGS-2200, Eyela Tokyo Rikakikai Co., Tokyo, Japan) at 38 °C. The concentrated lipid was methylated with 0.5 N NaOH (in methanol) and 14% boron trifluoride (in methanol) according to the method of David et al. [17]. After that, 5 mL of distilled water and 2 mL of hexane were mixed, centrifuged for 10 min at 2 °C and 3000× *g*, and 1 μL was injected into gas chromatography (CP-8400, Varian, Inc., Palo Alto, CA, USA) equipped with an HP-Innowax column (30 m length × 0.32 mm id × 0.25 μm film thickness, Agilent Technologies, Inc. Palo Alto, CA, USA) for analysis. At this time, the analysis conditions were inlet temperature: 260 °C, split ratio: 1/10, carrier: Heat 1 mL/min, oven program: 150 °C for 1 min, 150–200 °C at 15/min, 200–250 °C at 2/min, 250 °C for 10 min; FID temperature: 280 °C. Each analyzed fatty acid peak was calculated as a percentage (%) of the total fatty acid peak area after comparison and identification with the retention time of the standard material (47015-U, PUFA No. 2 Animal Source, Supelco, Bellefonte, PA, USA).

### 2.11. Statistical Analysis

The experimental results were analyzed after conducting at least three repetitions of the experiment. The statistical processing program SAS (version 9.3 for Windows, SAS Institute Inc., Cary, NC, USA) was used to test the significance of the experimental results, and the results were expressed as mean and standard error of the means (SEM). One-way ANOVA (Analysis of Variance) was performed on the experimental results following the general linear model (GLM) procedure, and the significance was verified at *p* < 0.05 using Duncan’s multiple range test.

All the sample sizes were 30 and the normality test was carried out by the Shapiro–Wilk test.

## 3. Results and Discussion

### 3.1. Proximate Composition and Shear Force

Table 1 shows the proximate composition and shear force according to the slaughter age of black goats.

As the slaughter age increased, the moisture content tended to decrease, whereas the fat content increased. The moisture content appeared to decrease because of the increase in fat and collagen contents. Arain et al. [18] reported that the fat content increased as the age of goats increased, which was similar to the results of this study. Collagen content increased significantly as the slaughter age increased (*p* < 0.05), which is consistent with the fact that the amount of collagen and the degree of complexity increases with increasing animal age and physical activity [13]. Hwang et al. [13] reported that the collagen content of black goat sirloin was significantly higher at 18 months of age than at 9 months of age, which was similar to the results of this study. 

Shear force tended to increase as the slaughter age of the black goat increased, and at 36 months, it showed the significantly highest value at 47.82 N (*p* < 0.05). The value of 47.82 N is about 4.88 kg, and Belew et al. [19] classified meat with a shear force exceeding 4.6 kg as “tough” as a degree of tenderness. Tenderness, one of the important characteristics consumers consider when ingesting meat, is affected by age, breed, and gender [20]. In addition, connective tissues such as collagen and elastin may have a negative effect on the tenderness of meat as the tissue structure becomes denser with age [21]. In addition, the presence of thermally stable collagen cross-links limits the solubility of collagen in meat even at high temperatures [22]. In this study, it was postulated that the increase in shear force value with increasing slaughter age could be caused by the increase in collagen cross-links.

### 3.2. Color and pH_24_

The color and pH_24_ of black goats according to the slaughter age are shown in Table 2. 

The lightness of black goats tended to decrease as the slaughter age increased. Abhijith et al. [22] reported that the heme iron content and lightness of meat have a negative correlation, and the decrease in lightness with increasing slaughter age seemed attributable to the increase in redness. Redness became significantly higher as the slaughter age increased (*p* < 0.05), which was thought to be caused by the increase in the number of type I muscle fibers and myoglobin content according to the increase in slaughter age. In general, the myoglobin content in the muscle is affected by the age of the animal and the type of muscle fiber [23], and it was observed that the lower the slaughter age, the lower the myoglobin content [24]. In addition, Bakhsh et al. [25] reported that the proportion of type Ⅰ muscle fibers in Korean native black goats was significantly higher at 18 months than at 9 months. Yellowness tended to increase as the slaughter age increased. Similarly, Zhang et al. [26] reported that the IMF (Intramuscular fat) content and yellowness of sheep showed a significant positive correlation. Therefore, it was concluded that the increase in yellowness according to the increase in slaughter age in this study is caused by the increase in fat content. 

The pH_24_ of black goats tended to increase with increasing slaughter age, and 12, 24, and 36 months of age showed a significantly higher value than those of 3, 6, and 9 months of age (*p* < 0.05). This is due to the increase in the number of type I muscle fibers and the increase in pH as the slaughter age increased [25,27]. The pH_24_ values, according to the slaughter ages analyzed in this study, ranged from 5.79 to 6.09, and goat meat has been reported to have a higher pH compared to the normal pH range of 5.4 to 5.7 for other edible meats [5]. The goat is known to be highly susceptible to perimortem stress, which limits muscle acidification post-mortem, resulting in a high pH in the goat [28]. Moreover, Kawęcka et al. [29] reported that goats at 12 months of age exhibited significantly higher pH than that of 9 months of age, showing similar results to this study. 

### 3.3. Water Holding Capacity and Cooking Yield

The water-holding capacity of meat is an indicator of improved juiciness and is an important factor in determining the functional properties and process suitability of meat products [30]. The water-holding capacity and cooking yield of black goats according to the slaughter age are shown in Figure 1, and the water-holding capacity tended to increase with slaughter age. 

This seemed to be caused by the increase in pH as the slaughter age increased. When the pH value deviates further from the isoelectric point, the negative charge increases, resulting in a repulsive force between the muscle filaments, which expands the space between the filaments, increasing the water-holding capacity [31]. In a study on the muscle fiber types of beef, Li et al. [32] also reported that the higher the content of type I muscle fibers, the higher the water-holding capacity. The cooking yield was significantly higher at 12, 24, and 36 months of age compared to 3, 6, and 9 months of age (*p* < 0.05), and there was no significant difference between 12, 24, and 36 months of age. The cooking yield is affected by the water-holding capacity of the meat, and a higher water-holding capacity is associated with a higher fixed water bound to the protein in the muscle, also leading to an increase in the cooking yield [33]. Saccà et al. [34] reported that the heating loss decreased with goat slaughter age, showing similar results to this study. Therefore, the results of the cooking yield experiment showed that the slaughter of black goats is appropriate after 12 months of age.

### 3.4. Minerals

Minerals are influenced by several factors, such as animal breed, sex, age, and diet. because minerals regulate human body functions, they are considered significant in terms of nutrition [35]. Table 3 shows the mineral content according to the slaughter age of black goats, and the P content showed no significant difference between the treatments.

The K and Na contents tended to increase as the slaughter age increased up to 12 months of age, but there was no significant difference between 12, 24, and 36 months of age. Both K and Na are important factors for muscle contraction and relaxation in humans, and as Na is an antagonist of K, it forms membrane potential on both sides of the cell membrane and performs the function of transmitting nerve impulses; therefore, in case of deficiency, symptoms such as muscle cramps may appear [36,37]. The Mg and Ca contents increased until 12 months, and thereafter, the contents decreased significantly compared to the 12 months of age. Schönfeldt et al. [38] reported that the Mg content increased and then decreased as the age of cattle increased in the mineral analysis study of beef sirloin, which was similar to the results of this study. In addition, in terms of human health, Ca contributes to the contraction and relaxation of blood vessels, muscle function, and bone development, and Mg is an important mineral factor for energy metabolism and protein synthesis [39]. As a result of mineral content analysis in this study, 12 months of age is the appropriate slaughter age in terms of Mg and Ca, and it was judged that it is appropriate to slaughter after 12 months of age in terms of Na and K. However, the reason why the mineral content decreases again was not clear, and various additional studies are needed.

### 3.5. Free Amino Acid

Black goats have a similar level of essential amino acid content to other livestock [40]. Because essential amino acids cannot be synthesized in the human body, black goat meat can be an excellent source of essential amino acids [41]. In addition, free amino acids are not only important as components of bioactive substances but also affect sensory characteristics of meat, such as taste and flavor [42]. The free amino acid composition of black goats according to the slaughter age is shown in Table 4, in which all free amino acids increased with slaughter age. Histidine, threonine, valine, and phenylalanine, which are essential amino acids, showed the highest content at the age of 36 months significantly (*p* < 0.05), and the contents of isoleucine, leucine, and tryptophan showed no significant difference between the ages of 12, 24, and 36 months. Generally, it is known that glutamine and aspartic acid represent umami taste, alanine, and glycine represent sweet taste, and arginine, valine, leucine, and phenylalanine represent bitter taste [43]. Glutamine and aspartic acid tended to increase as the slaughter age increased, but no significant difference was observed after 9 months of age. On the other hand, valine and phenylalanine showed significantly higher contents at 36 months of age than at other months (*p* < 0.05). An increase in free amino acids that show bitterness, such as valine, phenylalanine, tyrosine, and leucine, can have a negative impact on sensory properties because they offset desirable flavor components such as glutamine and glycine [44]. Total free amino acid contents tended to increase with slaughter age, and there was no significant difference between 12, 24, and 36 months. This seemed to be attributable to tyrosine because it accounted for the highest content. In this study, when looking at the flavor and total free amino acid content, it was judged that the slaughter of black goats is appropriate between 9 months and 24 months.

### 3.6. Fatty Acid Composition

Fatty acid composition in meat affects organoleptic properties and quality and plays an important role in providing essential nutrients to humans and maintaining a healthy body weight [6,45]. Table 5 shows the free fatty acid composition of black goat meat samples according to the age of slaughter. Oleic acid (C18:1 n9) and palmitic acid (C16:0) showed the highest contents at all slaughter ages, which are the same results as other fatty acid studies in black goat and sheep [6,46]. The contents of essential fatty acids, linoleic acid (C18:2n6) and arachidonic acid (C20:4n6), showed no significant difference between the slaughter ages, but the content of α-linoleic acid (C18:3n3) was significantly the highest at 12 months of age (*p* < 0.05). The saturated fatty acid (SFA) content showed a significantly higher value at the age of 3 months (53.62%) compared to other treatment groups (*p* < 0.05) and declined until the age of 12 months, followed by an increase. In a fatty acid analysis of goats according to slaughter age, Toplu et al. [47] reported that the SFA of sirloin slaughtered at 3 months of age was the significantly highest and it showed a high SFA value caused by the fatty acid composition of goat milk consumed by young goats. Additionally, Budimir et al. [48] reported that the SFA content increased with the slaughter age of sheep, showing similar results to this study. In this study, it is presumed that the SFA content decreased after weaning at 3 months of age and then increased again with age. Unsaturated fatty acid (UFA) content increased until 12 months of age and then decreased from 24 months, which is thought to be influenced by the SFA content. The SFA, mainly animal oil, raises blood cholesterol levels in our body and causes various metabolic diseases, whereas UFA is an essential fatty acid that increases HDL cholesterol levels and makes blood vessels healthy [49]. The PUFA/SFA ratio was 0.19% at 12 months of age, which was significantly higher than other slaughter ages (*p* < 0.05). It has been reported that ruminants showed a lower PUFA/SFA ratio than the recommended ratio of 0.4 to 0.5 caused by the biohydrogenation of PUFA, and beef (0.06~0.12) and lamb (0.15) showed low levels of PUFA/SFA ratio [50,51]. In addition, Toplu et al. [47] reported that the PUFA/SFA ratio was about 0.1 in the fatty acid analysis of goats according to the slaughter age, which was similar to the results of this study. Therefore, as a result of the fatty acid composition analysis in this study, it was judged that the optimal slaughter age of black goats is 9 to 12 months.

## 4. Conclusions

In this study, the physicochemical properties of black goat meat according to the slaughter age were investigated. In conclusion, the optimal slaughter age of black goats in this study was after 12 months for water holding capacity, cooking yield, and mineral content, 9 to 24 months for free amino acids, and 9 to 12 months for fatty acid composition. However, the results for free amino acids after 24 months of age and fatty acid composition at 3 months and after 24 months shows that the meat quality might not be fully suitable for consuming at these times. Furthermore, this study has limitations due to the small number of biological replicates, and more additional research must be carried out. Therefore, it is considered that this study can be used as basic data for establishing the optimal slaughter age of black goats and researching black goat meat products.

## Figures and Tables

**Figure 1 animals-13-00692-f001:**
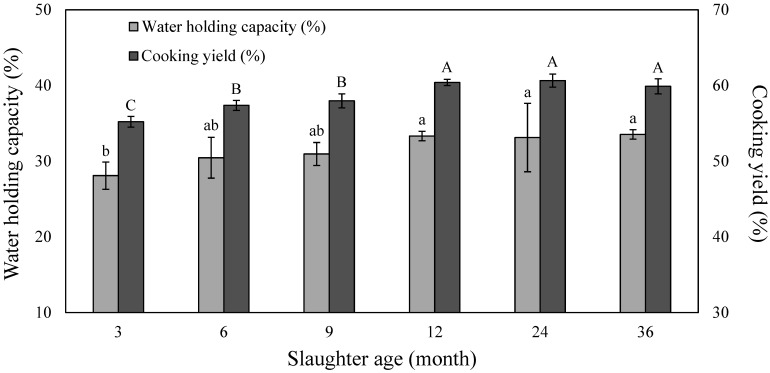
Water holding capacity and cooking yield of black Korean goat meat with different slaughter ages. ^a,b^ Means on the same bar with different letters are significantly different (*p* < 0.05). ^A–C^ Means on the same bar with different letters are significantly different (*p* < 0.05).

**Table 1 animals-13-00692-t001:** Proximate composition and shear force of black Korean goat meat with different slaughter ages.

Trait	Slaughter Age (Months)						SEM ^1^ (*n* = 30)
	3	6	9	12	24	36	
Moisture (%)	74.51 ^a^	73.61 ^b^	73.30 ^bc^	72.87 ^c^	71.02 ^d^	71.32 ^d^	0.06
Crude protein (%)	20.57 ^b^	21.62 ^a^	21.71 ^a^	21.28 ^a^	21.57 ^a^	21.42 ^a^	0.06
Crude fat (%)	3.29 ^b^	3.39 ^b^	3.51 ^b^	4.89 ^a^	5.64 ^a^	5.67 ^a^	0.08
Collagen (%)	1.15 ^f^	1.22 ^e^	1.29 ^d^	1.43 ^c^	1.52 ^b^	1.68 ^a^	0.01
Shear force (N)	25.81 ^d^	27.53 ^d^	37.70 ^c^	39.66 ^bc^	42.82 ^b^	47.82 ^a^	0.30

^a–f^ Mean values in the same row with different letters are significantly different (*p* < 0.05). ^1^ Standard error of the means.

**Table 2 animals-13-00692-t002:** Color and pH_24_ of black Korean goat meat with different slaughter ages.

Trait		Slaughter Age (Months)						SEM ^1^ (*n* = 30)
		3	6	9	12	24	36	
Color	CIE L*	44.99 ^a^	42.47 ^b^	37.15 ^c^	35.68 ^d^	35.43 ^d^	35.47 ^d^	0.10
	CIE a*	5.65 ^f^	7.44 ^e^	9.01 ^d^	11.00 ^c^	12.23 ^b^	14.03 ^a^	0.07
	CIE b*	5.23 ^e^	6.13 ^d^	6.50 ^d^	7.33 ^c^	9.17 ^b^	10.03 ^a^	0.07
pH_24_		5.79 ^c^	5.87 ^bc^	5.89 ^b^	6.06 ^a^	6.07 ^a^	6.09 ^a^	0.01

^a–f^ Mean values in the same row with different letters are significantly different (*p* < 0.05). ^1^ Standard error of the means.

**Table 3 animals-13-00692-t003:** Mineral composition of black Korean goat meat with different slaughter ages.

Trait (mg/100 g)	Slaughter Age (Months)						SEM ^1^ (*n* = 30)
	3	6	9	12	24	36	
P	195.06	193.86	194.91	203.46	197.11	198.61	0.89
K	14.85 ^c^	15.55 ^b^	14.99 ^c^	16.41 ^a^	16.22 ^a^	15.89 ^ab^	0.04
Mg	16.16 ^b^	16.43 ^b^	17.35 ^b^	22.23 ^a^	17.49 ^b^	17.34 ^b^	0.30
Ca	4.30 ^bc^	4.35 ^bc^	4.92 ^b^	5.84 ^a^	4.30 ^bc^	4.05 ^c^	0.06
Na	8.61 ^c^	9.14 ^c^	10.04 ^b^	10.86 ^a^	10.58 ^ab^	10.61 ^ab^	0.06

^a–c^ Mean values in the same row with different letters are significantly different (*p* < 0.05). ^1^ Standard error of the means.

**Table 4 animals-13-00692-t004:** Free amino acid of black Korean goat meat with different slaughter ages.

Trait (μ mole/g dw)	Slaughter Age (Months)						SEM ^1^ (*n* = 30)
	3	6	9	12	24	36	
Histidine	0.06 ^c^	0.08 ^c^	0.14 ^b^	0.17 ^b^	0.19 ^b^	0.25 ^a^	0.01
Asparagine	0.08 ^d^	0.12 ^cd^	0.20 ^bc^	0.21 ^b^	0.26 ^ab^	0.32 ^a^	0.01
Serine	0.24 ^b^	0.28 ^b^	0.59 ^a^	0.63 ^a^	0.65 ^a^	0.82 ^a^	0.02
Glutamine	2.06 ^b^	2.60 ^b^	2.91 ^ab^	2.93 ^ab^	3.18 ^ab^	3.80 ^a^	0.09
Arginine	1.95 ^c^	2.69 ^abc^	2.42 ^bc^	2.43 ^bc^	3.08 ^ab^	3.38 ^a^	0.09
Glycine	0.77 ^b^	0.86 ^ab^	1.11 ^ab^	1.20 ^ab^	1.27 ^a^	1.30 ^a^	0.05
Aspartic acid	0.05 ^b^	0.08 ^b^	0.21 ^a^	0.23 ^a^	0.27 ^a^	0.30 ^a^	0.01
Threonine	0.36 ^c^	0.39 ^c^	0.63 ^bc^	0.75 ^b^	0.85 ^b^	1.17 ^a^	0.03
Alanine	0.16 ^c^	0.17 ^c^	0.35 ^b^	0.43 ^ab^	0.50 ^ab^	0.60 ^a^	0.02
Cysteine	0.96 ^c^	1.20 ^c^	1.79 ^b^	1.87 ^ab^	2.09 ^ab^	2.44 ^a^	0.07
Tyrosine	8.26 ^c^	11.30 ^c^	18.31 ^b^	20.97 ^ab^	21.28 ^ab^	24.26 ^a^	0.55
Methionine	0.11 ^c^	0.13 ^c^	0.25 ^b^	0.29 ^b^	0.35 ^ab^	0.42 ^a^	0.01
Valine	0.06 ^c^	0.12 ^bc^	0.16 ^bc^	0.18 ^b^	0.18 ^b^	0.29 ^a^	0.01
Isoleucine	0.08 ^c^	0.12 ^c^	0.26 ^b^	0.29 ^ab^	0.30 ^ab^	0.39 ^a^	0.01
Leucine	0.17 ^c^	0.22 ^c^	0.50 ^b^	0.59 ^ab^	0.59 ^ab^	0.76 ^a^	0.02
Phenylalanine	0.08 ^d^	0.12 ^cd^	0.19 ^bc^	0.24 ^b^	0.24 ^b^	0.33 ^a^	0.01
Tryptophan	-	-	0.01 ^b^	0.01 ^ab^	0.02 ^a^	0.02 ^a^	0.01
Total	15.22 ^d^	20.47 ^c^	30.03 ^b^	33.42 ^ab^	35.29 ^ab^	40.86 ^a^	0.50

^a–d^ Mean values in the same row with different letters are significantly different (*p* < 0.05). ^1^ Standard error of the means.

**Table 5 animals-13-00692-t005:** Fatty acid composition of black Korean goat meat with different slaughter age.

Trait (%)	Slaughter Age (Months)						SEM ^1^ (*n* = 30)
	3	6	9	12	24	36	
Myristic acid (C14:0)	10.50 ^a^	5.30 ^b^	3.65 ^c^	2.31 ^c^	3.019 ^c^	2.52 ^c^	0.09
Palmitic acid (C16:0)	30.82 ^a^	27.35 ^b^	24.32 ^bc^	21.66 ^c^	27.19 ^b^	25.93 ^b^	0.26
Palmitoleic acid (C16:1n7)	2.21	2.06	2.13	1.75	2.06	1.95	0.06
Stearic acid (C18:0)	12.30 ^b^	15.84 ^ab^	13.78 ^ab^	14.98 ^ab^	16.11 ^ab^	17.04 ^a^	0.33
Oleic acid (C18:1n9)	36.35 ^c^	42.22 ^b^	49.39 ^a^	51.68 ^a^	46.99 ^ab^	47.83 ^a^	0.38
Vaccenic acid (C18:1n7)	0.08 ^a^	0.06 ^ab^	0.02 ^c^	0.02 ^bc^	0.06 ^a^	0.04 ^abc^	0.01
Linoleic acid (C18:2n6)	4.70	4.62	4.21	4.93	2.94	2.92	0.16
α-linolenic acid (C18:3n3)	0.14 ^b^	0.13 ^b^	0.25 ^b^	0.43 ^a^	0.17 ^b^	0.13 ^b^	0.01
Gondoic acid (C20:1n9)	0.09	0.08	0.06	0.05	0.08	0.07	0.01
Arachidonic acid (C20:4n6)	2.42	2.00	1.94	1.87	1.15	1.38	0.09
Eicosapentaenoic acid (C20:5n3)	0.10 ^ab^	0.08 ^ab^	0.11 ^ab^	0.17 ^a^	0.09 ^ab^	0.05 ^b^	0.01
Docosatetraenoate acid (C22:4 n6)	0.21	0.18	0.09	0.09	0.11	0.11	0.01
Docosahexaenoic acid (C22:6 n3)	0.06 ^a^	0.06 ^a^	0.04 ^ab^	0.04 ^ab^	0.02 ^b^	0.02 ^ab^	0.01
SFA	53.62 ^a^	48.49 ^b^	41.75 ^cd^	38.98 ^d^	46.31 ^bc^	45.49 ^bc^	0.33
UFA	46.37 ^d^	51.50 ^c^	58.23 ^ab^	61.02 ^a^	53.67 ^bc^	54.50 ^bc^	0.33
MUFA	38.73 ^c^	44.42 ^bc^	51.60 ^a^	53.50 ^a^	49.19 ^ab^	49.89 ^ab^	0.41
PUFA	7.64	7.08	6.63	7.52	4.48	4.61	0.23
n3	0.30 ^b^	0.28 ^b^	0.40 ^ab^	0.63 ^a^	0.28 ^b^	0.20 ^b^	0.02
n6	7.34	6.81	6.24	6.89	4.20	4.41	0.23
UFA/SFA	0.87 ^d^	1.06 ^cd^	1.40 ^ab^	1.57 ^a^	1.17 ^c^	1.20 ^bc^	0.02
MUFA/SFA	0.73 ^d^	0.92 ^cd^	1.24 ^ab^	1.37 ^a^	1.07 ^bc^	1.10 ^bc^	0.02
PUFA/SFA	0.14 ^ab^	0.15 ^ab^	0.16 ^ab^	0.19 ^a^	0.10 ^b^	0.10 ^b^	0.01
n6/n3	23.75	25.06	15.98	12.01	25.04	21.49	1.24

^a–d^ Mean values in the same row with different letters are significantly different (*p* < 0.05). ^1^ Standard error of the means.

## Data Availability

No publicly archived dataset for this experiment.

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
