# Peer review of "Physicochemical Properties of Black Korean Goat Meat with Various Slaughter Ages"

_animals, 2023, doi:10.3390/ani13040692_

Round 1
Reviewer 1 Report
The paper presents an interesting study, particularly because, as I understand it, it reinforces the importance of goats, which normally has little research on it, and in particular native breeds, such as the Black Korean goat. The writing is very well presented and written. However, there are details that must be attended.
1.L78. How long were the samples kept at 24 C? What analyzes were performed during that time? If the time is long, the process of oxidation of hemoglobin and fats begins, which can interfere with the results.
2.L100. How long after slaughter was the water holding capacity realized?
3.L108. At what temperature was the Cooking yield determined?
4.L58. Why did you use Duncan's test?...There are other more reliable ones.
5.L326. The conclusions the authors present are a general summary of results. I suggest to modify and conclude correctly.
6.L35-40. The first paragraph of the Introduction I do not find any relationship with the study carried out, at least the authors do not express it. I suggest removing it or justifying the relationship with the study.
7. The discussion can be strengthened if the authors take into consideration the aspect of the characteristics of the black Korean goat. It would be worth including this information to potentiate the importance of such a breed, and the contribution to universal access to knowledge.
Author Response
Thank you for reviewing this research spending your precious time.
We have reflected your opinions as much as possible.
Point 1: L78. How long were the samples kept at 24 C? What analyzes were performed during that time? If the time is long, the process of oxidation of hemoglobin and fats begins, which can interfere with the results.
Response 1: Samples were stored at -24°C to inhibit fat oxidation and used for the experiment while being stored for one week. pH, color, and water holding capacity were analyzed 24 hours after slaughter, and the rest of the experiments were also examined within the storage period
Point 2: L100. How long after slaughter was the water holding capacity realized?
Response 2: Water holding capacity was analyzed 24 hours after slaughter.
Point 3: L108. At what temperature was the Cooking yield determined?
Response 3: The cooking yield was measured after heating at 80°C for 30 minutes and then cooling at 25 °C for 1 hour. Accordingly, the cooking yield method was modified.
Point 4: L58. Why did you use Duncan's test?...There are other more reliable ones.
Response 4: We thought that Duncan's test would be good for animal experiments because it could actively reveal differences in treatment groups. In future studies, we will consider using other test methods based on your opinion.
Point 5: L326. The conclusions the authors present are a general summary of results. I suggest to modify and conclude correctly.
Response 5: Revised it to reflect the comments on the manuscript.
Point 6: L35-40. The first paragraph of the Introduction I do not find any relationship with the study carried out, at least the authors do not express it. I suggest removing it or justifying the relationship with the study.
Response 6: Removed the first paragraph of the introduction.
Reviewer 2 Report
This document demonstrates academic and scientific solvency by providing precise information in the area of meat science, specifically in goats. The way in which they focus on resolving a situation of global scope seems appropriate to me, considering the advantages of providing healthy and nutritionally favorable food. The document presents a complete analysis of the object of study. The results of the research are an example of linking science with social causes.
Author Response
Thank you for reviewing this research spending your precious time.
We have fully appreciated your comments.
May god bless you.
Reviewer 3 Report
The authors analyzed the physicochemical properties of black goat meat according to the slaughter age (3, 6, 9, 12, 24, 36 months). And the results showed that 12 months of age is the most nutritionally superior among various black goat slaughter days. Therefore, the authors concluded nutritionally excellent black goat slaughter age was occur at 12 months of age. The article is clearly and concisely written, and the structure is presented in a logical manner.
Specific comments follow:
1. 2.1 sample preparation: a total of 30 female black goats…, why the authors choose “female” goats, how about “male” goats? Is there difference in nutrition composition in various slaughter ages?
2. Were the 5 goats in each age group same breed? Were they fed with same feed and cultured in same circumstances?
3. The body weight, carcass yield and body fat rate of the goats should be shown.
Author Response
Thank you for reviewing this research spending your precious time.
We have reflected your opinions as much as possible.
Point 1: 2.1 sample preparation: a total of 30 female black goats…, why the authors choose “female” goats, how about “male” goats? Is there difference in nutrition composition in various slaughter ages?
Response 1: From an industrial point of view, males are bred for seed preservation reasons, while females produce by-products such as goat milk and have a smaller unique odor than males, so they are widely bred for meat. Therefore, we decided to use females, which are widely used for meat, in the experiment.
Point 2: Were the 5 goats in each age group same breed? Were they fed with same feed and cultured in same circumstances?
Response 2: Revised it to reflect the comments on the manuscript.
Point 3: The body weight, carcass yield and body fat rate of the goats should be shown.
Response 3: Revised it to reflect the comments on the manuscript.
Reviewer 4 Report
This study compared the meat quality traits of Korean black goat from different slaughter ages. This reviewer has several concerns about this study:
1. Referencing in this study is a concern, especially in the introduction, this reviewer has spot quite a few cited references failed to support the statement. The authors needed to make sure the references were cited properly and not abused.
2. Only 5 biological replicates were used for animal study with no information about the animal on-farms, as the authors observed that the diet could be an important factor affecting the meat composition.
3. Materials and methods were too brief, lacking a lot of key information. Also the measurement for fatty acids was total fatty acids not the free form.
4. Results and discussion: discussions regarding free amino acids were wrong as only free amino acids were measured in this study. Overall the discussions were superficial lacking in-dept interpretation of the results and explanations.
5. Conclusion read like another abstract and needed to be re-written.
Other specific comments:
L34-44: This reviewer was not convinced by the argument provided here about the goat consumption and senior-friendly concept. As the authors described in the next paragraph, goat meat has a unique flavour, which is the main issue for consumer acceptance,
L35-L37: This sentence needs to be revised as poor English. Also reference No 1 is a paper in Korean language, I was expecting one from UN, could the authors provide the original reference for this statement?
L44-45: As the statement her was related to consumer interest and trend for food choice, one should expect the following references were based on consumer studies, while the two references provided here seemed to be indirect citations from a review on sheep/goat, and a study on soybean substitution for chicken breast?? Please revise.
L46: Bovine or Bovidae?
L48-49: what are the extracts? Please specify.
L52: ref 9 – the study cited here did not support the statement for consumers changed their preference to meat.
L54-56: the stats referenced here described the global goat meat consumption stats in their study, not for black goat.
L62: goat parts or meat cuts?
L74-75: Expecting more information regarding those goats – how many farms were they from? What kinds of feeds did they have on farm? How were they slaughtered?
L80-82: please provide more details regarding how the samples were prepared, and the procedures for analyses. What were the principle for measuring the proximate composition? Based on AOAC methods?
L86: what force? Trigger force?
L96-99: How the pH of meat was measured using a flat surface pH probe?
L100: how many replicates were taken for WHC measurement?
L108: how the samples were cooked, to what temperature? What kind of samples were used? Please, provide more details throughout the whole methodology section.
L112: what minerals were measured; did you use any standards to calculate the concentration? More details are required here.
L125-127: poor English
L134: So only 4 amino acids were detected and quantified?
L137: Were the free fatty acids measured or total fatty acids? Lipid were extracted and methylated.
L156-157: the standard mix provided here was a polyunsaturated fatty acid mix, I assumed only PUFA was measured here?
L173-174: if the changes were significant, particularly for fat content, there was no need to use “tended”.
L176: Goat vs. cattle, I don’t think this was an appropriate comparison.
L178-179: I think the crosslinking of connective tissues decrease the solubility of collagen, not the reason for increase of collagen with age.
L182: please remove “tended to”.
L183: why using “kg” when reporting newton in the table? So 4.88kg is still considered tender?
L206: I don’t think using a beef patties was appropriate to explain the relationship between yellowness and fat content as subcutaneous fat was involved, while for the goat sample in this study was mainly IMF.
L214: Ref 34 - this particular study did not report the optimum pH range for goat. Please check and revise. Could the authors provide a basis for why the pH of goat meat was higher than other meats, and the older ones were higher than then young ones?
L221-222: Fig 1 – the cooking losses for those samples were about 40% or more, which were quite large, the cooking methods must be included in the methodology.
L255-257, L261-262: Were these functions related to goat or human health?
L268-269: Need references to support the statement.
L268-270: Please be careful when making these statements as this study only measured free amino acids, not the total amino acids. The same issue was found for the rest of this section, thus all the discussions in this section were wrong and should be re-written.
L289: Table 4: only L-glutamine, L-asparagine, L-tryptophan, γ-aminobutyric acid were used as standard, how did the authors measure other amino acids?
L296-297: Poor English
L309-310: if diet was considered as an effect, the detailed information regarding how these goats from different ages were fed was required for this study.
L314-315: Poor English
L321-322: what is the basis for the fatty acid variations due to slaughter ages?
L323: Table 5 – only PUFA standards were used in this study, how did the authors measure the other SFA and MUFA? Also why presented as % of total fatty acids, not the absolute among of fatty acids in meat?
L326: The conclusion section reads like another abstract, the authors should focus on drawing a conclusion from the data obtained in this study not just simply summarising it again. Also any suggestions for future studies?
Author Response
Thank you for reviewing this research spending your precious time.
We have reflected your opinions as much as possible.
Point 1: L34-44: This reviewer was not convinced by the argument provided here about the goat consumption and senior-friendly concept. As the authors described in the next paragraph, goat meat has a unique flavour, which is the main issue for consumer acceptance,
Response 1: Removed the first paragraph of the introduction.
Point 2: L35-L37: This sentence needs to be revised as poor English. Also reference No 1 is a paper in Korean language, I was expecting one from UN, could the authors provide the original reference for this statement?
Response 2: Removed the first paragraph of the introduction.
Point 3: L44-45: As the statement her was related to consumer interest and trend for food choice, one should expect the following references were based on consumer studies, while the two references provided here seemed to be indirect citations from a review on sheep/goat, and a study on soybean substitution for chicken breast?? Please revise.
Response 3: Removed the first paragraph of the introduction.
Point 4: L46: Bovine or Bovidae?
Response 4: Revised it to reflect the comments on the manuscript.
Point 5: L48-49: what are the extracts? Please specify.
Response 5: Revised it to reflect the comments on the manuscript.
Point 6: L52: ref 9 – the study cited here did not support the statement for consumers changed their preference to meat.
Response 6: Revised it to reflect the comments on the manuscript.
Point 7: L54-56: the stats referenced here described the global goat meat consumption stats in their study, not for black goat.
Response 7: Revised it to reflect the comments on the manuscript.
Point 8: L62: goat parts or meat cuts?
Response 8: Revised it to reflect the comments on the manuscript.
Point 9: L74-75: Expecting more information regarding those goats – how many farms were they from? What kinds of feeds did they have on farm? How were they slaughtered?
Response 9: Revised it to reflect the comments on the manuscript.
Point 10: L80-82: please provide more details regarding how the samples were prepared, and the procedures for analyses. What were the principle for measuring the proximate composition? Based on AOAC methods?
Response 10: Revised it to reflect the comments on the manuscript.
Point 11: L86: what force? Trigger force?
Response 11: Revised it to reflect the comments on the manuscript.
Point 12: L96-99: How the pH of meat was measured using a flat surface pH probe?
Response 12: Revised it to reflect the comments on the manuscript.
Point 13: L100: how many replicates were taken for WHC measurement?
Response 13: More than three experiments were conducted, which were explained in statistical analysis.
Point 14: L108: how the samples were cooked, to what temperature? What kind of samples were used? Please, provide more details throughout the whole methodology section.
Response 14: Revised it to reflect the comments on the manuscript.
Point 15: L112: what minerals were measured; did you use any standards to calculate the concentration? More details are required here.
Response 15: Revised it to reflect the comments on the manuscript.
Point 16: L125-127: poor English
Response 16: Revised it to reflect the comments on the manuscript.
Point 17: L134: So only 4 amino acids were detected and quantified?
Response 17: On article, the ‘amino acid standard’ was used to measure the remaining amino acids.
Point 18: L137: Were the free fatty acids measured or total fatty acids? Lipid were extracted and methylated.
Response 18: Revised it to reflect the comments on the manuscript.
Point 19: L156-157: the standard mix provided here was a polyunsaturated fatty acid mix, I assumed only PUFA was measured here?
Response 19: The fatty acid composition was determined using PUFA standards and references are follow:
- https://doi.org/10.1016/j.meatsci.2017.08.004
- https://www.sigmaaldrich.com/KR/ko/product/supelco/47015u
Point 20: L173-174: if the changes were significant, particularly for fat content, there was no need to use “tended”.
Response 20: Revised it to reflect the comments on the manuscript.
Point 21: L176: Goat vs. cattle, I don’t think this was an appropriate comparison.
Response 21: Revised it to reflect the comments on the manuscript.
Point 22: L178-179: I think the crosslinking of connective tissues decrease the solubility of collagen, not the reason for increase of collagen with age.
Response 22: Revised it to reflect the comments on the manuscript.
Point 23: L182: please remove “tended to”.
Response 23: Revised it to reflect the comments on the manuscript.
Point 24: L183: why using “kg” when reporting newton in the table? So 4.88kg is still considered tender?
Response 24: This is our description error and revised it to reflect the comments on the manuscript.
Point 25: L206: I don’t think using a beef patties was appropriate to explain the relationship between yellowness and fat content as subcutaneous fat was involved, while for the goat sample in this study was mainly IMF.
Response 25: Revised it to reflect the comments on the manuscript.
Point 26: L214: Ref 34 - this particular study did not report the optimum pH range for goat. Please check and revise. Could the authors provide a basis for why the pH of goat meat was higher than other meats, and the older ones were higher than then young ones?
Response 26: Revised it to reflect the comments on the manuscript.
Point 27: L221-222: Fig 1 – the cooking losses for those samples were about 40% or more, which were quite large, the cooking methods must be included in the methodology.
Response 27: Revised it to reflect the comments on the manuscript.
Point 28: L255-257, L261-262: Were these functions related to goat or human health?
Response 28: Revised it to reflect the comments on the manuscript.
Point 29: L268-269: Need references to support the statement.
Response 29: Revised it to reflect the comments on the manuscript.
Point 30: L268-270: Please be careful when making these statements as this study only measured free amino acids, not the total amino acids. The same issue was found for the rest of this section, thus all the discussions in this section were wrong and should be re-written.
Response 30: Revised it to reflect the comments on the manuscript.
Point 31: L289: Table 4: only L-glutamine, L-asparagine, L-tryptophan, γ-aminobutyric acid were used as standard, how did the authors measure other amino acids?
Response 31: On article, the ‘amino acid standard’ was used to measure the remaining amino acids.
Point 32: L296-297: Poor English
Response 32: Revised it to reflect the comments on the manuscript.
Point 33: L309-310: if diet was considered as an effect, the detailed information regarding how these goats from different ages were fed was required for this study.
Response 33: Revised it to reflect the comments on the manuscript.
Point 34: L314-315: Poor English
Response 34: Revised it to reflect the comments on the manuscript.
Point 35: L321-322: what is the basis for the fatty acid variations due to slaughter ages?
Response 35: Changes in fatty acids due to slaughter age may have resulted from an increase in unsaturated fatty acid content as fat content increased, which needs to be identified through future research. And, revised it to reflect the comments on the manuscript.
Point 36: L323: Table 5 – only PUFA standards were used in this study, how did the authors measure the other SFA and MUFA? Also why presented as % of total fatty acids, not the absolute among of fatty acids in meat?
Response 36: The fatty acid content was measured with the PUFA standard, and if you check the product specifications of this standard, all 13 fatty acids in this study can be measured. Also, it is because the data that came out as a peak through GC was expressed as a percentage of the area.
Point 37: L326: The conclusion section reads like another abstract, the authors should focus on drawing a conclusion from the data obtained in this study not just simply summarising it again. Also any suggestions for future studies?
Response 37: Revised it to reflect the comments on the manuscript. In addition, there are plans to study black goats in the future.

Reviewer 5 Report
Dear all,
In this manuscript, Da-Mi Choi et al. assessed to determine Physicochemical properties of black Korean goat meat with various slaughter ages.
The information of manuscript has a significance for choosing human consumption of goat meat. However, the experimental design is not clear, owing to the following reasons:
A total of 30 female back goats were collected from the Gaon. But:
Did these goats raise by the same farmer?
Were their mothers the same age? how many mothers?
Total of experimental goats?
How to choose goats for slaughter?
…
Therefore, in my opinion, this manuscript is currently unacceptable for review. I’m not saying it can’t be in the future, but it needs a lot of work to meet Animals standards.
Author Response
Thank you for reviewing this research spending your precious time.
We have reflected your opinions as much as possible.
Point 1: Did these goats raise by the same farmer?
Response 1: Black goats were bred on the same farm, and revised it to reflect the comments on the manuscript.
Point 2: Were their mothers the same age? how many mothers?
Response 2: A total of 30 offspring were obtained from 20 mother black goats, and the average age of the mother black goats was 1 to 2 years.
Point 3: Total of experimental goats?
Response 3: A total of 30 black goats were used.
Point 4: How to choose goats for slaughter?
Response 4: Selecting the slaughter age was modified by the research of Toplu et al., and an additional 24-month-old and 36-month-old were added to further supplement the analysis.
- ‘https://doi.org/10.1007/s11250-013-0360-0’
Reviewer 6 Report
General comment:
The aim of the paper is interesting and falls within the scope of the Journal. However, a number of aspects need attention. In my opinion, much information in the materials and methods section is lacking or obscure, and the conclusions are not fully supported by the results. Please see also the specific comments:
Lines 43-44: The relationship between elderly people and meat goat consumption is not completely clear to me. Did you mean to say that the consumption of low-fat, high-protein meat is increasing only in the elderly?
Line 46: “bovine” did you mean “bovidae”?
Lines 54-56: Please report some data on the black goat diffusion, it is important to understand the international relevance of the paper
Lines 62-65: I do not think this part is appropriate, the paper focused on fresh meat rather than processed meat.
Lines 66-67: Please report in the introduction section the studies on the optimal slaughter age of black goats already carried out.
Line 74: The animals were Boer x black Korean goats. Please clarify in text this choice.
Lines 74-75: In my opinion, there is a lot of information missing that is very important for the understanding of the results (e.g. how were animals reared? number of pen, Were animals reared indoor or outdoor? What is the available area per animal in the pen? How were the characteristics of the diets? Were all animals slaughtered on the same days? Were the goats in the 6 experimental groups (one trial) reared at the same time? How were the goats slaughtered? And so on)
Line 158: Five animals per experimental group seems a rather low number to achieve an adequate power of the statistical test adopted. Please report the statistical power of ANOVA performed.
Line 159: It is not clear what do you mean by “after a minimum of three repeated trials”. Do you mean that three replications were performed for each analysis? Please clarify it.
Line 162: Did you check Anova assumptions (normality, homoscedasticity,…)? Please report in text the test used.
Results and discussion section: to understand the results, it is necessary to report at least the slaughter weight per experimental group.
Tables: Please add n=…, in each Table
Lines 187-188: Please consider that the meat tenderness is usually related to the insolubility of collagen rather than its content.
Line 240: “because goat slaughtered at 12 months of age showed the optimal cooking yield”, please consider that the cooking yield was similar at 12-24 and 36 months.
Lines 263-266: I do not agree, this statement is true only for Mg and Ca, not for Na and K.
Lines 268-269: Please add an appropriate reference for “Black goats have a higher content of essential amino acids than other livestock species”
Lines 286-288: I do not agree, it seems that, in general, the highest amino acids contents were found at 36 months (e.g. lines 272-274), and the total amino acids content was similar at 12-24 and 36 months.
Chapter 3.5 and 3.6: Please improve the discussion by trying to explain the reasons for the differences obtained.
Table 5. In my opinion, the unit of measurement is not clear, you report %, do you mean % of the total identified fatty acids?
Lines 321-322: The main significant differences concern C18:3n-3, but in terms of total amount and, therefore, from a nutritional point of view, the differences between experimental groups in this fatty acid seem very low. Considering the aim of the study, the authors should consider this aspect as well.
Lines 335-337: “Such results show that 12 months of age should be optimal nutritionally among various black goat slaughter ages”. I do not think the conclusions are supported by the data; the choice of meat from 12 month old goat is not supported by proximate composition (e.g. crude fat), collagen content, shear force, color, and amino acids, but only by Mg, Ca and C18:3n-3.
Author Response
Thank you for reviewing this research spending your precious time.
We have reflected your opinions as much as possible.
Point 1: Lines 43-44: The relationship between elderly people and meat goat consumption is not completely clear to me. Did you mean to say that the consumption of low-fat, high-protein meat is increasing only in the elderly?
Response 1: Removed the first paragraph of the introduction.
Point 2: Line 46: “bovine” did you mean “bovidae”?
Response 2: Revised it to reflect the comments on the manuscript.
Point 3: Lines 54-56: Please report some data on the black goat diffusion, it is important to understand the international relevance of the paper
Response 3: Revised it to reflect the comments on the manuscript.
Point 4: Lines 62-65: I do not think this part is appropriate, the paper focused on fresh meat rather than processed meat.
Response 4: Revised it to reflect the comments on the manuscript.
Point 5: Lines 66-67: Please report in the introduction section the studies on the optimal slaughter age of black goats already carried out.
Response 5: Revised it to reflect the comments on the manuscript.
Point 6: Line 74: The animals were Boer x black Korean goats. Please clarify in text this choice.
Response 6: The reason why we choose this breed is that the crossbreed between the black Korean goat and the Boer goat is the most widely bred and distributed in South Korea.
Point 7: Lines 74-75: In my opinion, there is a lot of information missing that is very important for the understanding of the results (e.g. how were animals reared? number of pen, Were animals reared indoor or outdoor? What is the available area per animal in the pen? How were the characteristics of the diets? Were all animals slaughtered on the same days? Were the goats in the 6 experimental groups (one trial) reared at the same time? How were the goats slaughtered? And so on)
Response 7: Revised it to reflect the comments on the manuscript.
Point 8: Line 158: Five animals per experimental group seems a rather low number to achieve an adequate power of the statistical test adopted. Please report the statistical power of ANOVA performed.
Response 8: As your request, we are providing some statistical power and normality of the data. CIE L* (statistical power = 0.960, normality = 0.0005), Mineral Mg (statistical power = 0.824, normality = 0.1302), Fatty acid composition n6/n3 (statistical power = 0.488, normality = 0.6190), Proximate composition Crude protein (statistical power = 0.838, normality = 0.0928)
Point 9: Line 159: It is not clear what do you mean by “after a minimum of three repeated trials”. Do you mean that three replications were performed for each analysis? Please clarify it.
Response 9: Revised it to reflect the comments on the manuscript.
Point 10: Line 162: Did you check Anova assumptions (normality, homoscedasticity,…)? Please report in text the test used.
Response 10: As your request, we are providing some statistical power and normality of the data. CIE L* (statistical power = 0.960, normality = 0.0005), Mineral Mg (statistical power = 0.824, normality = 0.1302), Fatty acid composition n6/n3 (statistical power = 0.488, normality = 0.6190), Proximate composition Crude protein (statistical power = 0.838, normality = 0.0928)
Point 11: Results and discussion section: to understand the results, it is necessary to report at least the slaughter weight per experimental group.
Response 11: Revised it to reflect the comments on the manuscript.
Point 12: Tables: Please add n=…, in each Table
Response 12: Revised it to reflect the comments on the manuscript.
Point 13: Lines 187-188: Please consider that the meat tenderness is usually related to the insolubility of collagen rather than its content.
Response 13: Revised it to reflect the comments on the manuscript.
Point 14: Line 240: “because goat slaughtered at 12 months of age showed the optimal cooking yield”, please consider that the cooking yield was similar at 12-24 and 36 months.
Response 14: Revised it to reflect the comments on the manuscript.
Point 15: Lines 263-266: I do not agree, this statement is true only for Mg and Ca, not for Na and K
Response 15: Revised it to reflect the comments on the manuscript.
Point 16: Lines 268-269: Please add an appropriate reference for “Black goats have a higher content of essential amino acids than other livestock species”
Response 16: Revised it to reflect the comments on the manuscript.
Point 17: Lines 286-288: I do not agree, it seems that, in general, the highest amino acids contents were found at 36 months (e.g. lines 272-274), and the total amino acids content was similar at 12-24 and 36 months.
Response 17: Revised it to reflect the comments on the manuscript.
Point 18: Chapter 3.5 and 3.6: Please improve the discussion by trying to explain the reasons for the differences obtained.
Response 18: Revised it to reflect the comments on the manuscript.
Point 19: Table 5. In my opinion, the unit of measurement is not clear, you report %, do you mean % of the total identified fatty acids?
Response 19: Due to the data obtained as peaks through GC, the fatty acid compositions are expressed as a percentage of the area.
Point 20: Lines 321-322: The main significant differences concern C18:3n-3, but in terms of total amount and, therefore, from a nutritional point of view, the differences between experimental groups in this fatty acid seem very low. Considering the aim of the study, the authors should consider this aspect as well.
Response 20: Revised it to reflect the comments on the manuscript.
Point 21: Lines 335-337: “Such results show that 12 months of age should be optimal nutritionally among various black goat slaughter ages”. I do not think the conclusions are supported by the data; the choice of meat from 12 month old goat is not supported by proximate composition (e.g. crude fat), collagen content, shear force, color, and amino acids, but only by Mg, Ca and C18:3n-3.
Response 21: Revised it to reflect the comments on the manuscript.
Round 2
Reviewer 4 Report
This manuscript has been improved, while some revisions are still required as follows:
Abstract: where were the results for colour, water holding capacity and shear force?
L23: this was not true as the cook yield remained the same for 12, 24, and 36 months old goat.
L24: if the change was significant, pleas remove the “tendency”. The same to the other results.
L46: What year?
L82: how the samples were prepared for shear force measurement and how the samples were sheared when measuring?
L85: ok, now it became a problem, trigger force of 5.6N seemed to be high, please double check if this is right.
L95-97: I don’t think this is an appropriate way to measure the pH of meat.
L108-110: the thickness of samples for cooking? Are they consistent in thickness? Is it cooked in a bag or what? Why the current cooking method was chosen? It is still obscure.
L218-219: inappropriate statement and citation – ref 27 was about different muscle types, not slaughter age. Ref 25 was about slaughter age and mentioned fiber types, while there was no linkage between pH and type I muscle fiber.
L222: “post-stress”? it didn’t make sense.
L265-266: what could be the basis for the increase of minerals with slaughter age to 12 months then decrease?
L273-294: Please revise the whole section and remove all the discussion related to nutrition as free amino acids were measured here, not total amino acids. Free amino acids were released during post-mortem degradation of protein, while only limited levels of proteins will be hydrolysed and tuned into free amino acids, thus highlighting nutritional benefit is not appropriate. Something may be worth considering that free amino acids are commonly linked with flavour of meat, not the nutritional benefit.
L298: subtitle 3.6 was not revised. Please remove “Free” as total fatty acids was measured here.
L314-315: this sentence is confusing – the difference in SFA due to slaughter age was caused by slaughter age?? Regarding diet, it this is considered as one of the main factors, please elaborate and explain how the feeding used in this study could affect the fatty acids.
L320: please remove “%” and also in the following sentences, ratio is not a percentage.
L326-330: results presented in this study did not support this statement, the fat content increased with age, but the unsaturated fatty acids increase only up to 12 months and decrease afterwards. Also, why more than 12 months is beneficial?
L338-340: this statement is confusing, why the decrease of mineral and fatty acids after 12 months old was recommended?
Author Response
Thank you for reviewing this research spending your precious time.
We have reflected your opinions as much as possible, so please check the attached file.
Response to Reviewer 4 Comments
Point 1: Abstract: where were the results for colour, water holding capacity and shear force?
Response 1: Revised it to reflect the comments on the manuscript.
Point 2: L23: this was not true as the cook yield remained the same for 12, 24, and 36 months old goat.
Response 2: Revised it to reflect the comments on the manuscript.
Point 3: L24: if the change was significant, pleas remove the “tendency”. The same to the other results.
Response 3: Since no significant differences were found among all age groups, the term trend was used.
Point 4: L46: What year?
Response 4: Revised it to reflect the comments on the manuscript.
Point 5: L82: how the samples were prepared for shear force measurement and how the samples were sheared when measuring?
Response 5: Revised it to reflect the comments on the manuscript.
Point 6: L85: ok, now it became a problem, trigger force of 5.6N seemed to be high, please double check if this is right.
Response 6: Revised it to reflect the comments on the manuscript.
Point 7: L95-97: I don’t think this is an appropriate way to measure the pH of meat.
Response 7: As such, there is a study that measured pH in the same way as ours, and please refer to the research that used this method. Therefore, we consider that our pH measurement method can be used.
https://doi.org/10.1016/j.afres.2022.100259
Point 8: L62: L108-110: the thickness of samples for cooking? Are they consistent in thickness? Is it cooked in a bag or what? Why the current cooking method was chosen? It is still obscure.
Response 8: We have been using this method in general and please refer to the research that used this method.
https://doi.org/10.3390/ani11092565
https://doi.org/10.3390/foods10092152
Point 9: L218-219: inappropriate statement and citation – ref 27 was about different muscle types, not slaughter age. Ref 25 was about slaughter age and mentioned fiber types, while there was no linkage between pH and type I muscle fiber.
Response 9: References have been added.
Point 10: L222: “post-stress”? it didn’t make sense.
Response 10: Revised it to reflect the comments on the manuscript.
Point 11: L265-266: what could be the basis for the increase of minerals with slaughter age to 12 months then decrease?
Response 11: The cause is unclear, and this opinion has been added to the manuscript.
Point 12: L273-294: Please revise the whole section and remove all the discussion related to nutrition as free amino acids were measured here, not total amino acids. Free amino acids were released during post-mortem degradation of protein, while only limited levels of proteins will be hydrolysed and tuned into free amino acids, thus highlighting nutritional benefit is not appropriate. Something may be worth considering that free amino acids are commonly linked with flavour of meat, not the nutritional benefit.
Response 12: Revised it to reflect the comments on the manuscript.
Point 13: L298: subtitle 3.6 was not revised. Please remove “Free” as total fatty acids was measured here.
Response 13: Revised it to reflect the comments on the manuscript.
Point 14: L314-315: this sentence is confusing – the difference in SFA due to slaughter age was caused by slaughter age?? Regarding diet, it this is considered as one of the main factors, please elaborate and explain how the feeding used in this study could affect the fatty acids.
Response 14: Revised it to reflect the comments on the manuscript.
Point 15: L320: please remove “%” and also in the following sentences, ratio is not a percentage.
Response 15: Revised it to reflect the comments on the manuscript.
Point 16: L326-330: results presented in this study did not support this statement, the fat content increased with age, but the unsaturated fatty acids increase only up to 12 months and decrease afterwards. Also, why more than 12 months is beneficial?
Response 16: This is our descriptive error, we have corrected it.
Point 17: L338-340: this statement is confusing, why the decrease of mineral and fatty acids after 12 months old was recommended?
Response 17: This is our descriptive error, we have corrected it.
Reviewer 5 Report
Dear all,
In the revised manuscript, Da-Mi Choi et al. added some information on materials and methods, also other parts of the manuscript.
However, some following questions should be clearly explained:
1) According to my knowledge, you had slaughtered a total of 30 black Korean female goats at 5 different times (6 ones per one time). However, you said that all slaughtered goats from 20 mothers having ages ranging from 1 to 2 years old. You did not clearly explain how to choose 6 female goats each time (For example 3 goats from their mothers at 1-year-old and 3 ones at 2-year-old; or slaughtered goat weight at each time was the average weight of the experimental goat herd or other ones?)
2) Feeds and their nutritional composition at each developed stage of goat are missing. Besides, uncontrolled feed intake may influence studied indicators.
3) Monitoring goats were born from many different mothers, statistical treatment with one-way ANOVA is not satisfied.
4) Slaughtered goats at 3 months (weaning) for studying are not a reality
Therefore, in my opinion, the revised manuscript is not enough strong to persuade for reviewing. It needs a lot of work to meet Animals standards.
Author Response
Thank you for spending your precious time to reviewing this research .
We have read your opinions and we will revise your suggestions on next paper.
We are thank you for your concerns again.
Reviewer 6 Report
The revised manuscript has partially took care of the reviewer's suggestions. I have to report some comments that were not entirely satisfied:
Material and methods section: Since the paper focused on meat quality, it is important to report at least the energy and protein content/concentration of the animals’ diet.
Statistical analysis section: Please also report in the text at least the statistical test used to assess the normality of data distribution
Lines 271-272: “Therefore, as a result of the mineral content experiment, it is considered that the slaughter of black goats is appropriate after 12 months of age”. I do not agree, P is statistically similar across all the experimental groups, K is similar at 6 and 36months, Ca is similar at 6 and 24-36-3 months, Na is similar at 9, 24 and 36 months
Lines 293-294: “it is considered that the slaughter of black goats is appropriate after 12 months of age “, I do not agree, the highest level of total aa was found at 36 months as well the highest content of Histidine, threonine, valine, and phenyl-alanine. The content of aa and total aa was statistically similar at 9 and 12 months
Lines 329-330: “In conclusion, as a result of fatty acid analysis, it is judged that the slaughter 329 age of black goats is more than 12 months”, I do not agree , e.g. the FA profile is similar at 9 and 12 months with the only exception of C18:3n-3, (total n3content is also statistically similar between 9 and 12 months)
Lines 336-340: “The results of water holding capacity, cooking yield, mineral content, and fatty acid composition demonstrate that the optimal slaughter age for black goats is 12 months” . “In particular, since the content of minerals and fatty acids composition decreases after 12 months of age”. As I previously explained and also in my previous report, these conclusions are not supported by results (your statement is true only for Mg, Ca, and C18:3n-3).
Author Response
Thank you for reviewing this research spending your precious time.
We have reflected your opinions as much as possible, so please check the attached file.
Response to Reviewer 6 Comments
Point 1: Material and methods section: Since the paper focused on meat quality, it is important to report at least the energy and protein content/concentration of the animals’ diet.
Response 1: Revised it to reflect the comments on the manuscript.
Point 2: Statistical analysis section: Please also report in the text at least the statistical test used to assess the normality of data distribution
Response 2: Revised it to reflect the comments on the manuscript.
Point 3: Lines 271-272: “Therefore, as a result of the mineral content experiment, it is considered that the slaughter of black goats is appropriate after 12 months of age”. I do not agree, P is statistically similar across all the experimental groups, K is similar at 6 and 36months, Ca is similar at 6 and 24-36-3 months, Na is similar at 9, 24 and 36 months
Response 3: Revised it to reflect the comments on the manuscript.
Point 4: L46: Lines 293-294: “it is considered that the slaughter of black goats is appropriate after 12 months of age “, I do not agree, the highest level of total aa was found at 36 months as well the highest content of Histidine, threonine, valine, and phenyl-alanine. The content of aa and total aa was statistically similar at 9 and 12 months
Response 4: Revised it to reflect the comments on the manuscript.
Point 5: Lines 329-330: “In conclusion, as a result of fatty acid analysis, it is judged that the slaughter 329 age of black goats is more than 12 months”, I do not agree , e.g. the FA profile is similar at 9 and 12 months with the only exception of C18:3n-3, (total n3content is also statistically similar between 9 and 12 months)
Response 5: Revised it to reflect the comments on the manuscript.
Point 6: Lines 336-340: “The results of water holding capacity, cooking yield, mineral content, and fatty acid composition demonstrate that the optimal slaughter age for black goats is 12 months” . “In particular, since the content of minerals and fatty acids composition decreases after 12 months of age”. As I previously explained and also in my previous report, these conclusions are not supported by results (your statement is true only for Mg, Ca, and C18:3n-3).
Response 6: Revised it to reflect the comments on the manuscript.
Round 3
Reviewer 4 Report
Overall the manuscript has been improved, and some minor comments as follows:
L16: As the age of the black goats increased….
L21-22, L25: if it’s insignificant, then indicated that P>0.05.
Abstract: please revise the whole abstract and reduce the repetition, if attributes changed in a similar way with slaughter age, put them together in one sentence.
L31: slaughter age, not slaughter days.
L31-32: the last sentence is the same as the one (L29-31) before. Please choose one to keep.
L53-54: What was the main outcome of this study?
L86: What did you mean by “heating in parallel to the muscle fiber direction”?
L173: This should be included in result section not here. Also, what did the normality results tell the reader? Some descriptions and comments are needed.
Tables: I think the P-values result tables. Also, if the results showed no significant differences, no letter should be used. Currently almost all the results included letters indicating they were significantly differed from one another. However, according to authors’ previous responses to point 3, some results were not significant? If that’s right then why they were discriminated by different letters? As stated in the footnote of the tables, different letters suggested significant difference?
Table 5: Please double check your letters and SEM. Most SEM are quite low, while how come the 12 months (33.42) was similar to 36 months (40.86)?
L305: cancel desirable flavour? Poor English, please revise.
Table 6: As the results were shown as % fatty acids, should the combination of SFA and UFA equals to 100%, also the statistical difference (letters) should be the same for both SFA and UFA. However it was not the case here. Please check your stats. Also, some large differences in PUFA and n-6 with the increase of age, while the results were insignificant indicating large variations in the results, likely due to the low number of animal replicates (n=5).
Conclusion: I think the low number of biological replicates should be pointed out in the conclusion as the limitation of this study.
Author Response
Thank you for reviewing this research spending your precious time.
We have reflected your opinions as much as possible, so please check the attached file.
The content of attached file:
Point 1: L16: As the age of the black goats increased….
Response 1: Revised it to reflect the comments on the manuscript.
Point 2: L21-22, L25: if it’s insignificant, then indicated that P>0.05.
Response 2: Each experiment has a significant difference between the treatment sections, and we used the word 'tend' because it is an overall explanation. Therefore, I think it is appropriate not to mark p>0.05.
Point 3: Abstract: please revise the whole abstract and reduce the repetition, if attributes changed in a similar way with slaughter age, put them together in one sentence.
Response 3: Revised it to reflect the comments on the manuscript.
Point 4: L31: slaughter age, not slaughter days.
Response 4: Revised it to reflect the comments on the manuscript.
Point 5: L31-32: the last sentence is the same as the one (L29-31) before. Please choose one to keep.
Response 5: Revised it to reflect the comments on the manuscript.
Point 6: L53-54: What was the main outcome of this study?
Response 6: Revised it to reflect the comments on the manuscript.
Point 7: L86: What did you mean by “heating in parallel to the muscle fiber direction”?
Response 7: The text has been amended to reflect your opinion.
Point 8: L173: This should be included in result section not here. Also, what did the normality results tell the reader? Some descriptions and comments are needed.
Response 8: It has been removed based on the opinions of other reviewers.
Point 9: Tables: I think the P-values result tables. Also, if the results showed no significant differences, no letter should be used. Currently almost all the results included letters indicating they were significantly differed from one another. However, according to authors’ previous responses to point 3, some results were not significant? If that’s right then why they were discriminated by different letters? As stated in the footnote of the tables, different letters suggested significant difference?
Response 9: It has been removed based on the opinions of other reviewers.
Point 10: Table 5: Please double check your letters and SEM. Most SEM are quite low, while how come the 12 months (33.42) was similar to 36 months (40.86)?
Response 10: Errors were checked and corrected.
Point 11: L305: cancel desirable flavour? Poor English, please revise.
Response 11: Revised it to reflect the comments on the manuscript.
Point 12: Table 6: As the results were shown as % fatty acids, should the combination of SFA and UFA equals to 100%, also the statistical difference (letters) should be the same for both SFA and UFA. However it was not the case here. Please check your stats. Also, some large differences in PUFA and n-6 with the increase of age, while the results were insignificant indicating large variations in the results, likely due to the low number of animal replicates (n=5).
Response 12: As a result of fatty acid analysis, the level of γ-Linolenic acid was 0~0.2%, so low that we did not include γ-Linolenic acid in the data. Therefore, it seems that the sum of the fatty acid results is not 100%.
Point 13: Conclusion: I think the low number of biological replicates should be pointed out in the conclusion as the limitation of this study.
Response 13: Revised it to reflect the comments on the manuscript.
Reviewer 6 Report
The Authors revised the manuscript considering the reviewer's suggestions, but some doubts remain:
-My previous comment: “Statistical analysis section: Please also report in the text at least the statistical test used to assess the normality of data distribution”
Answer of the Authors: “Revised it to reflect the comments on the manuscript”
The Authors reported the test used to assess the normality of data distribution as recommended and I thank the Authors for this. Usually, to choose the statistical test in a one-way ANOVA with 6 levels, the normality of the residuals of the model should be tested. If the normality of residuals was tested in Table 1, I am worry because for some variables the ANOVA assumption is violated and this test is not appropriate. If in Table 1 the Authors simply report the results for the distribution of each variable, in my opinion, having 6 groups, it makes no sense. In any case, in my opinion, Table 1 should be deleted.
-Lines 353-355: “In particular, as a result of amino acid and fatty acid analysis, slaughter after 24 months of age may have a negative effect on the meat quality of black goats, so care must be taken”, I agree for amino acids, but not for fatty acids in terms of SFA, MUFA, UFA, n-3 (e.g. 36 months is similar to 6 months,…).
Author Response
Thank you for reviewing this research spending your precious time.
We have reflected your opinions as much as possible, so please check the attached file.
The content of the attached file:
Point 1: The Authors reported the test used to assess the normality of data distribution as recommended and I thank the Authors for this. Usually, to choose the statistical test in a one-way ANOVA with 6 levels, the normality of the residuals of the model should be tested. If the normality of residuals was tested in Table 1, I am worry because for some variables the ANOVA assumption is violated and this test is not appropriate. If in Table 1 the Authors simply report the results for the distribution of each variable, in my opinion, having 6 groups, it makes no sense. In any case, in my opinion, Table 1 should be deleted.
Response 1: Revised it to reflect the comments on the manuscript.
Point 2: Lines 353-355: “In particular, as a result of amino acid and fatty acid analysis, slaughter after 24 months of age may have a negative effect on the meat quality of black goats, so care must be taken”, I agree for amino acids, but not for fatty acids in terms of SFA, MUFA, UFA, n-3 (e.g. 36 months is similar to 6 months,…).
Response 2: Revised it to reflect the comments on the manuscript.